# Neuroinflammation and Amyotrophic Lateral Sclerosis: Recent Advances in Anti-Inflammatory Cytokines as Therapeutic Strategies

**DOI:** 10.3390/ijms26083854

**Published:** 2025-04-18

**Authors:** Costanza Stacchiotti, Simona Mazzella di Regnella, Miriam Cinotti, Alida Spalloni, Elisabetta Volpe

**Affiliations:** 1Molecular Neuroimmunology Unit, Santa Lucia Foundation, 00143 Rome, Italy; c.stacchiotti@hsantalucia.it (C.S.); s.mazzella@hsantalucia.it (S.M.d.R.); m.cinotti@hsantalucia.it (M.C.); e.volpe@hsantalucia.it (E.V.); 2Department of Experimental Medicine, University of Rome Tor Vergata, 00133 Rome, Italy; 3Department of Biology and Biotechnology Charles Darwin, Sapienza University, 00185 Rome, Italy; 4Molecular Neurobiology Unit, Santa Lucia Foundation, 00143 Rome, Italy

**Keywords:** amyotrophic lateral sclerosis, neuroinflammation, cytokines

## Abstract

Neuroinflammation is an inflammatory response occurring within the central nervous system (CNS). The process is marked by the production of pro-inflammatory cytokines, chemokines, small-molecule messengers, and reactive oxygen species. Microglia and astrocytes are primarily involved in this process, while endothelial cells and infiltrating blood cells contribute to neuroinflammation when the blood–brain barrier (BBB) is damaged. Neuroinflammation is increasingly recognized as a pathological hallmark of several neurological diseases, including amyotrophic lateral sclerosis (ALS), and is closely linked to neurodegeneration, another key feature of ALS. In fact, neurodegeneration is a pathological trigger for inflammation, and neuroinflammation, in turn, contributes to motor neuron (MN) degeneration through the induction of synaptic dysfunction, neuronal death, and inhibition of neurogenesis. Importantly, resolution of acute inflammation is crucial for avoiding chronic inflammation and tissue destruction. Inflammatory processes are mediated by soluble factors known as cytokines, which are involved in both promoting and inhibiting inflammation. Cytokines with anti-inflammatory properties may exert protective roles in neuroinflammatory diseases, including ALS. In particular, interleukin (IL)-10, transforming growth factor (TGF)-β, IL-4, IL-13, and IL-9 have been shown to exert an anti-inflammatory role in the CNS. Other recently emerging immune regulatory cytokines in the CNS include IL-35, IL-25, IL-37, and IL-27. This review describes the current understanding of neuroinflammation in ALS and highlights recent advances in the role of anti-inflammatory cytokines within CNS with a particular focus on their potential therapeutic applications in ALS. Furthermore, we discuss current therapeutic strategies aimed at enhancing the anti-inflammatory response to modulate neuroinflammation in this disease.

## 1. Introduction

Neuroinflammation is a complex process that arises within the central nervous system (CNS) in response to injury, infections, autoimmunity or neurodegenerative diseases. The aim of this response is to protect the CNS by limiting the damage and facilitating the removal of pathogens, debris, and misfolded proteins. However, in some neurological diseases, the chronic inflammatory environment becomes detrimental and even contributes to neurodegeneration [1]. In this context, increasing evidence indicates that neuroinflammation contributes to motor neuron (MN) death, and influences the rate of disease progression in ALS [2,3]. The contributors to neuroinflammation can be broadly categorized into two main groups: resident CNS cells and infiltrating leukocytes, with their relative contributions varying among different kinds of CNS diseases [4]. In the CNS, astrocytes and microglia are the glial cells most responsive to changes in the microenvironment, capable of transitioning into a pro-inflammatory state. As the CNS damage progresses, glial cells release pro-inflammatory molecules to recruit peripheral blood cells, which can cross the damaged blood brain barrier (BBB), enter the CNS, and enhance neuroinflammation. During chronic neuroinflammation, glial cells may also produce neurotoxic factors which contribute to neurodegeneration. This mechanism has been largely investigated in multiple sclerosis (MS), characterized by a severely-compromised BBB, where infiltrating inflammatory monocytes, T lymphocytes, and B lymphocytes play a crucial role. In recent years, the view has emerged that brain barriers are not impermeable gates, but rather regulatory ones [5], and the disruption of this barrier control, which occurs in many neurological diseases [6], can lead to pathological leukocyte invasion from the periphery, thereby contributing to neuroinflammation in neurodegenerative diseases, including ALS.

However, a broad range of biological mechanisms are involved in the resolution of inflammation to restore tissue homeostasis and prevent the development of chronic inflammatory diseases.

In the CNS, glial cells and immune cells infiltrating the CNS possess the ability to release factors with anti-inflammatory properties, which can control the neuroinflammatory process by promoting tissue healing and homeostasis [7,8,9].

In this review, we discuss the contribution of anti-inflammatory cytokines in the resolution of neuroinflammation, highlighting the discoveries in preclinical and clinical studies and examining the potential therapeutic implications of engaging these molecules in ALS.

### Neuroinflammation in ALS

ALS is a fatal CNS neurodegenerative disease, characterized by degeneration of MN. Several mechanisms contribute to MN degeneration in ALS, such as oxidative damage, formation of intracellular protein aggregates, damage to axonal transport systems, mitochondrial dysfunction, impairment of RNA metabolism and of DNA damage repair, and excessive activation of glutamate receptors leading to neurotoxicity [10]. In addition, increasing evidence indicates that also neuroinflammation contributes to MN injury and progression of the disease [2,3,11]. Inflammation in the CNS is mediated by microglia, astroglia, and infiltrating immune cells. The involvement of non-neuronal cell types in ALS pathology introduced the concept of non-cell autonomous components contributing to ALS disease [12] (Figure 1). In particular, most studies indicate that inflammatory cytokines released by astrocytes and microglia trigger an irreversible pathological process that leads to the non-cell autonomous death of MN in patients with ALS [13]. Moreover, a compromised BBB has been described in ALS mouse models and patients with ALS [14], suggesting that entry of activated peripheral immune cells into the CNS parenchyma of patients with ALS might contribute to the neuroinflammatory process (Figure 1).

Interestingly, immune dysregulation characterized by microglial cell activation in the spinal cord of ALS patients [15], elevated biomarkers of neuro-inflammation in the cerebrospinal fluid (CSF) of ALS patients correlating with disease severity [16], and pro-inflammatory gene profiles of blood monocytes from ALS patients [17] were associated with decreased blood levels of a subset of T lymphocytes [2,18], named T regulatory (Treg) cells, known to act as negative regulators of inflammation [19]. Importantly, reduced levels of Treg cells in the blood have been shown to correlate with disease progression at the time of collection and to serve as predictors of future rapid progression [20,21].

Central and peripheral inflammatory mechanisms are important contributors to ALS both in the context of familiar [22,23,24] and sporadic disease [25,26]. In the context of familiar ALS, it has been reported that mutations in genes such as superoxide dismutase 1 (SOD1) and TAR DNA-binding protein 43 (TARDBP) promote microglia activation [27] through nuclear factor-kappa B (NF-κB) and NLRP3 pathways [28], while mutations in chromosome 9 open reading frame 72 (C9orf72) [23], TANK binding kinase 1 (TBK1) [29], and optineurin (OPTN) [30] affect the inflammatory type I interferon response.

Overall, although it is still unclear what initiates the inflammatory processes, especially in patients with sporadic ALS, neuroinflammation has an important role in ALS pathogenesis, and its neutralization is crucial to curb excessive tissue damage and promote repair processes during neurodegeneration.

## 2. Anti-Inflammatory Cytokines in the CNS

Among the endogenous mechanisms for countering inflammation, immune molecules known as anti-inflammatory cytokines play a major role. Cytokines, including interferons, interleukins, and chemokines, are molecules canonically associated with the peripheral immune system and regulating the peripheral inflammatory response. However, numerous studies have now demonstrated that many of these cytokines and their receptors are expressed by resident CNS cells or by CNS-infiltrating immune cells. Increasing evidence indicates that some cytokines within the CNS may have an anti-inflammatory role by limiting the neuroinflammatory responses (Figure 2). These anti-inflammatory cytokines might effectively inhibit neuroinflammation, either by affecting CNS infiltrating immune cells (for example, by skewing macrophage polarization, by enhancing Treg cells, or by interfering with the activation of infiltrating B or T lymphocytes), and by reducing activation of CNS resident microglia and astrocytes.

Among anti-inflammatory cytokines, IL-10 and TGF-β were the first to be described. They are both produced by Treg cells, known to suppress pathological immune responses in the settings of transplantation, allergy and autoimmune diseases [31]. Other cytokines known to govern the regulation and resolution of inflammation are those produced by T helper 2 (Th2) cells, such as IL-4 and IL-13 [32,33]. Similarly, IL-9, originally associated with Th2 cells and then identified as a specific cytokine of a new Th subtype named Th9 [34], is emerging as potential regulator of inflammatory processes. Finally, other cytokines are emerging as potentially relevant in inhibiting neuroinflammation, such as IL-35, IL-25, IL-37, and IL-27 [35,36,37,38].

### 2.1. Interleukin-10 in the CNS and in ALS

IL-10 was first discovered for its ability to inhibit activation and effector functions of pro-inflammatory cells. IL-10 acts through a functional receptor complex composed of two subunits, IL-10R1 (IL-10Rα) and IL-10R2 (IL-10Rβ), and mainly activates Janus kinase (Jak) 1/Tyrosine (Tyk) kinase 2, and is the signal transducer and activator of the transcription 3 (STAT3) system [39]. IL-10 is expressed by many immune cell types and plays a critical role in balancing immune responses to limit chronic inflammatory diseases [40]. In particular, IL-10 inhibits the production of several inflammatory cytokines, such as tumor necrosis factor (TNF)-α, IL-1β, IL-6, and interferon (IFN)-γ secretion from monocytes/macrophages [41,42]. Moreover, IL-10 is a powerful inhibitor of the generation of reactive oxygen species (ROS), and increases the release of TNF receptors, which may antagonize the effects of TNF-α [40,42]. In the course of CNS pathology, IL-10 limits neuroinflammatory processes, similarly to what happens in peripheral sites, influencing resident macrophages in order to contain their inflammatory responses and promote the mechanisms that maintain tissue integrity.

The neuroprotective role of IL-10 was also described in MS, where the production of IL-10 by a rare subset of B lymphocytes called regulatory B (Breg) cells was associated with immunosuppressive functions [43,44]. In line with this, in experimental autoimmune encephalomyelitis (EAE), a murine demyelinating and paralyzing model of MS in IL-10-deficient animals develop a more severe disease, while IL-10-overexpressing mice are highly resistant to EAE [45]. Consistently, in a mouse model of Parkinson’s disease (PD), IL-10 gene therapy has been shown to elicit immunomodulation with enhanced suppression of neuroinflammation associated with dopaminergic neuron survival [46].

The role of IL-10 in ALS is emerging as protective. First, clinical studies report an increase of IL-10 levels in the blood of ALS patients compared to control subjects [47,48], and then higher levels of IL-10 in the blood predict longer disease duration in ALS patients [49]. Moreover, the beneficial role of IL-10 is confirmed by Furukawa et al., which showed a positive correlation between IL-10 levels in the CSF of ALS patients and Revised ALS Functional Rating Scale (ALSFRS-R) scores, indicating that high IL-10 is associated with low disease progression [50]. In transgenic mice mutants for SOD1^G93A^, which model the clinicopathology of ALS, overexpression of IL-10 in the spinal cord through recombinant adeno-associated virus (rAAV) vectors has been associated with extended survival of mice [51,52]. Similarly, gene-therapy-induced IL-10 overexpression in microglia of SOD1^G93A^ mice significantly delays disease onset and increases survival [53]. In addition, a recent study demonstrated that intramuscular administration of IL-10 ameliorates disease progression and extends survival in SOD1^G93A^ [54]. Consistently, overexpression of IL-10 in the same ALS mouse model reduces inflammatory genes involved in innate immunity [51,52], as well as astrocytosis and microgliosis in the lumbar spinal cord, while the blocking of IL-10 exacerbates inflammation [53] (Figure 3).

### 2.2. Transforming Growth Factor-β in the CNS and in ALS

Another molecule known to exert anti-inflammatory functions is TGF-β. TGF-β binds the TGF-β receptor, which signals through a serine/threonine kinase domain that phosphorylates the signaling mediator Smad. TGF-β is produced by several immune and non-immune cell types, but the primary source of TGF-β1, which is the most common isoform, are Treg cells [55]. A crucial role of TGF-β was shown by its ability to block nuclear factor-B activation and pro-inflammatory cytokines [56].

TGF-β has been associated with the amelioration of neurotoxicity by reducing glial activation, particularly the microglial pro-inflammatory phenotype. It also decreases the expression of pro-inflammatory factors produced by glial and immune cells, such as TNF-α, inducible nitric oxide synthase (iNOS), IFN-γ, and IL-1β, while promoting the differentiation of Treg cells and the production of the anti-inflammatory cytokine IL-10 [57]. However, in addition to the regulatory effects of TGF-β, it should be noted that TGF-β can also promote Th17 differentiation and IL-17 production [58,59] in inflamed tissues, such as brain tissue in MS patients [60]. These opposing actions may generate contradictory results regarding the therapeutic potential of TGF-β in MS [61].

In ALS, the expression of the genes encoding TGF-β and components of the TGF-β signaling pathways are altered in the spinal cord and skeletal muscle of ALS mice, as well as in muscle biopsies from sporadic ALS patients [62,63], suggesting an involvement in ALS pathogenesis. Notably, the expression of TGFβ-RII expression correlates with the progression of reactive astrogliosis [62], and astrocyte-derived TGF-β1 has been shown to accelerate disease progression in ALS mice [64]. Furthermore, a selective inhibitor of TGFBRI kinase activity extends animal survival in the ALS mouse model, leading to the hypothesis that TGF-β1 may act as a negative prognostic factor, rather than a protective one [64]. In ALS patients, elevated TGF-β1 levels in serum, along with a trend towards increased TGF-β1 in spinal cord tissues, have been associated with a potential anti-inflammatory protective effect in the early stages of the disease, and with a neurotoxic effect in later disease [65]. Consistently, a dual role of TGF-β1 in ALS disease progression is further supported in a transient zebrafish model, where knockdown of *tgfb1a* partially prevented motor axon abnormalities and locomotor deficits [66] (Figure 3).

### 2.3. Interleukin-4 in the CNS and in ALS

IL-4 is a cytokine whose anti-inflammatory properties were discovered through its ability to suppress TNF-α and IL-1β production in lipopolysaccharide (LPS)-activated human monocytes [67]. Since then, the list of anti-inflammatory properties of IL-4 has grown, and IL-4 has also been shown to induce the production of the IL-1 receptor antagonist (IL-1ra) [68] and a decoy type II receptor [69], as well as to reduce IFN-γ production by promoting GATA-3 expression [70]. IL-4 receptor complexes (type 1 and type 2) mediate IL-4 signaling through the activation of Jak pathways and STAT6 phosphorylation [69]. The type I IL-4 receptor consists of IL-4 receptor alpha (IL-4Rα) and common γ chain (γc), while the type II IL-4 receptor comprises IL-4Rα and IL-13Rα1.

IL-4 is produced by eosinophils, basophils, mast cells, innate lymphoid T cells (ILC)2, natural killer (NK) T cells, and Th2 cells [71]. It plays a key role in type 2 immune responses, contributing to resistance against helminth parasites, inactivating toxins by stimulating B cell antibody production, and promoting the expansion of eosinophils, basophils, and mast cells [72]. Beyond its immune functions, IL-4 plays a pivotal anti-inflammatory role in the CNS by modulating the microglial polarization. IL-4 facilitates the transition from M1 microglia, the pro-inflammatory phenotype, to M2 microglia, the anti-inflammatory phenotype. This shift is characterized by a reduction in M1 markers (IL-1β, IL-6, TNF-α) and an increase in M2 markers (Arginase-1, TGF-β, IL-10, and CD206) [73]. Yang et al. demonstrated that early intracerebral injection of IL-4 in a rat model of intracerebral hemorrhage (ICH) promotes neurofunctional recovery by enhancing M2 activation and improving neurobehavioral outcomes [74]. The beneficial effects of IL-4 have also been highlighted in MS. In the EAE mouse model of MS, CNS-specific IL-4 deficiency resulted in disease exacerbation, accompanied by increased infiltration of inflammatory cells [75]. Moreover, several studies have shown that IL-4 treatment can reverse disease progression in EAE models [76,77,78]. These findings underscore the critical role of IL-4 as a key regulator of neuroinflammation within the CNS.

In SOD1^G93A^ mice, IL-4 gene therapy resulted in a general improvement in clinical outcomes during the early, slow progressive phase of the disease [79]. Additionally, Zhao et al. demonstrated that in the same model, IL-4 mediates the suppressive functions of Treg cells, including the inhibition of microglial cytotoxicity and the proliferation of T effector cells [80]. Interestingly, this suppressive effect on T cell effector proliferation is also mediated by other known anti-inflammatory cytokines, such as IL-10 and TGF-β [80]. Consistent with preclinical studies, Furukawa et al. reported increased IL-4 levels in the CSF of ALS patients compared to controls, as well as a negative correlation between IL-4 levels and disease progression in ALS patients [50]. In contrast, IL-4 levels in the CSF of ALS patients were negatively correlated with both the ALSFRS-R total and the bulbar subscales, indicating that the role IL-4 in ALS still remains unclear [81] (Figure 3).

### 2.4. Interleukin-13 in the CNS and in ALS

IL-13 shares many properties with IL-4, and it was initially identified as a T cell-derived cytokine that inhibits the production of inflammatory cytokines [82]. For this reason, it is typically classified as an anti-inflammatory interleukin. Like IL-4, IL-13 is secreted by Th2 cells, ILC2, NK T cells, as well as by mast cells, basophils, and eosinophils [83].

IL-13 has two receptors: the IL-4 type II receptor, composed of IL-4Rα and IL-13Rα1 chains, and IL-13Rα2 [84]; it shares a common signaling pathway with IL-4 [85]. IL-13 exerts similar functions to IL-4, including the regulation of M2 marker expression, such as the mannose receptor, Arginase-1, and IL-4Rα2, all of which are characteristic of anti-inflammatory macrophages [86,87]. In the context of CNS trauma IL-13 administration, either directly or through cell therapy, promotes an anti-inflammatory microglia/macrophage phenotype, and enhances the phagocytosis of damaged neurons in the peri-lesion areas, thereby providing neuroprotection [88,89,90,91,92]. The neuroprotective role of IL-13 has been further demonstrated by Li et al., who showed that IL-13 is upregulated in the brain and CSF of traumatic brain injury patients, with implications for synaptic plasticity and protection of neurons from excitotoxic death [93].

In addition, IL-13 has been reported to modulate brain inflammation by inducing the death of activated microglia, both in vitro and in vivo [94,95]. This effect is mediated by the enhancement of cyclooxygenase-2 (COX-2) expression, and by the production of PGE (2) and 15-deoxy-Delta(12,14)-PGJ(2) (15d-PGJ(2)), key factors involved in the resolution of brain inflammation [96]. In MS, the anti-inflammatory role of IL-13 has been widely reported. In an MS rat model, IL-13 reduced IL-1β and TNF expression as well as oxidative stress in macrophages, thereby mitigating disease severity [97]. In the cuprizone MS mouse model [98], IL-13 gene therapy conferred protection by enhancing the M2 phenotype of microglia and macrophages, thus limiting disease progression [99]. In EAE, treatment with human adipose tissue-derived stem cells (hADSCs) overexpressing IL-11 and IL-13 led to a reduction in the infiltration of inflammatory cells into the spinal cord and improved clinical outcomes [100]. In line with these findings, levels of IL-13 in the CSF of MS patients are positively correlated with axonal and neuronal integrity as well as with better performance in Multiple Sclerosis Functional Composite (MSFC) scoring [101].

Although the anti-inflammatory role of IL-13 in CNS has been largely demonstrated, its role in ALS has not been investigated. However, clinical data indicate an increase of IL-13 levels in the CSF of ALS patients compared to healthy subjects [81] and an increase of IL-13-producing T cells in the peripheral blood of ALS patients compared to healthy donors [102]. Importantly, the levels of IL-13 in CSF of ALS patients are inversely correlated with disease progression [103], while the percentage of IL-13-producing T cells in blood of ALS patients negatively correlates with the revised ALS functional rating scale scores, and positively correlates with the rate of disease progression [102]. These findings suggest that the precise role of IL-13 in ALS disease remains unclear (Figure 3).

**Figure 3 ijms-26-03854-f003:**
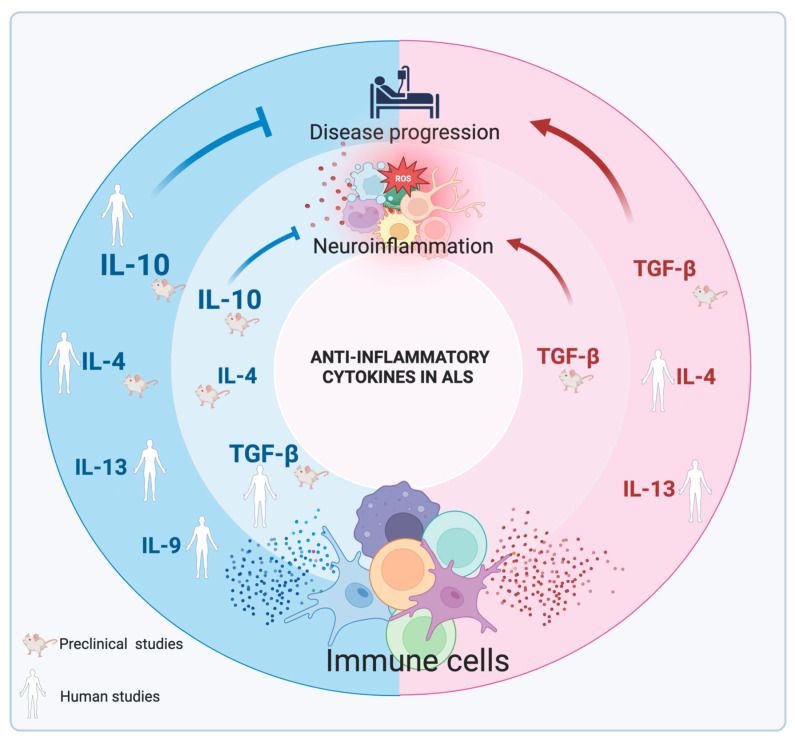
The role of anti-inflammatory cytokines in amyotrophic lateral sclerosis (ALS). Schematic representation of the anti-inflammatory cytokines’ effects in ALS, based on preclinical data in ALS animal models and correlative studies based on ALS patient-derived samples and their clinical information (specific references are reported). Arrows indicate the impact of cytokine on neuroinflammation and/or disease progression: sharp arrows represent exacerbation (pathogenic role), and blunt arrows represent inhibition (protective role). The protective effect of IL-10 is supported by different studies, and the protective role of IL-9 is predicted by a single study. In contrast, controversial results are reported about the roles of TGF-β, IL-4, and IL-13. Created in https://BioRender.com. Refs. [45,46,47,48,49,50,58,60,61,62,75,76,77,98,99].

### 2.5. Interleukin-9 in the CNS and in ALS

In recent years, the anti-inflammatory role of IL-9 has emerged. Initially associated with Th2 cells [104,105], IL-9 is now recognized as a defining feature of a specific subset of IL-9-producing CD4^+^ T cells, known as Th9 cells [34,106]. Interestingly, the anti-inflammatory cytokines TGF-β and IL-4 drive naive CD4^+^ T cells towards the Th9 profile [34,107], suggesting a regulatory function for IL-9.

IL-9 exerts its action through binding with its specific receptor (IL-9R), composed of the common γ chain, shared by several cytokine receptors, and the specific α-chain (IL-9Rα). IL-9/IL-9R binding triggers Jak kinases activation and phosphorylation of STAT-1, STAT-3, and STAT-5 [108], or activation of the PI3K/Akt and MAPK/ERK pathways [109,110]. IL-9 is involved in the expansion of mast cell populations during allergic responses [111] and in asthma [112,113]. Moreover, IL-9 is involved in immune responses against intestinal nematodes, promoting parasite expulsion from the gut [114,115]. The anti-inflammatory properties of IL-9 were first demonstrated by its ability to reduce respiratory burst and inflammatory cytokine release in human monocytes and alveolar macrophages [116,117]. In line with this, human macrophages stimulated with IL-9 exhibit reduced expression of inflammatory markers and increased TGF-β production, suggesting that IL-9 decreases their activation state and enhances their anti-inflammatory properties [118]. This anti-inflammatory role was also observed in the brain of MS patients, where IL-9 expression was inversely correlated with macrophage/microglia activation [118]. Consistently, in the CSF of MS patients, IL-9 levels inversely correlate with neurofilament levels, a marker of neurodegeneration, as well as with levels of IL-17, which are indicative of neuroinflammation [119]. In line with these results, mice lacking IL-9R exhibit an increase in IL-17-producing cells during EAE, which is associated with enhanced disease progression [120]. Moreover, systemic and local administration of IL-9 in EAE reduces clinical disability by decreasing TNF release by microglia [121]. In contrast, studies using IL-9 neutralization and IL-9 receptor deficiency in the EAE model showed contrasting results [122,123], likely due to experimental differences that warrant further investigation. However, there are no studies on the role of IL-9 in ALS. To date, clinical data have reported an increase of IL-9 levels in the CSF of ALS patients compared to control subjects [81], and that IL-9 higher levels in the CSF predict longer survival in ALS patients [49], suggesting that IL-9 may play protective role in ALS (Figure 3).

### 2.6. Emerging Anti-Inflammatory Cytokines and Their Role in the CNS

In recent years, the anti-inflammatory roles of newly discovered cytokines such as IL-35, IL-25, IL-37, and IL-27 have emerged.

IL-35 was discovered in 2007 and is highly expressed by cells with immunosuppressive properties, including Treg cells [124], Breg cells [125,126], tolerogenic dendritic cells (tolDCs) [127], and M2 macrophages [128]. IL-35 is composed of the IL-12α chain (p35) and the IL-27β chain Epstein-Barr virus-induced gene 3 (Ebi3), and binds to its receptor IL-35R, which consists of the IL-12Rβ2 and gp130 chains. The binding of IL-35/IL-35R activates the Jak signaling cascade, leading to the phosphorylation of transcriptional activators STAT1, STAT4, and STAT3 [129]. Importantly, IL-35 has been shown to play a significant role in the CNS. For instance, IL-35 alleviates inflammatory responses induced by oxidative stress in a brain stroke model [128], and in hypoxic-ischemic encephalopathy [130]. In addition, in EAE, IL-35 attenuates neuroinflammation by inducing IL-10 production, ultimately reducing disease progression [36,125].

IL-25, also known as IL-17E, is a member of the IL-17 cytokine family with recognized anti-inflammatory properties [131]. To elicit its functional responses, IL-25 requires the presence of both IL-17RA and IL-17RB [132]. It is produced by activated Th2 cells, mast cells, eosinophils, and alveolar macrophages [133,134,135,136,137], and is known for its role in enhancing allergic inflammation by amplifying Th2 immune responses [133,138]. However, in recent years, several anti-inflammatory properties of IL-25 have been identified, including its ability to suppress IL-17-producing T cells [139], inhibit the release of inflammatory cytokines by activated monocytes [140], and induce the polarization of M2 macrophages [141]. These findings have led to the investigation into the effects of IL-25 in EAE, where Th17 cells and inflammatory macrophages play a key pathogenic role. Recent studies have revealed that IL-25 treatment reduces neuroinflammation and EAE disease severity [37,139], suggesting that IL-25 and IL-17, despite being members of the same cytokine family, play opposing roles in the pathogenesis of MS. Similar findings have been observed in mouse models of neuroinflammation, such as those involving the entorhinal cortex lesions [37].

IL-37, discovered in 2000, is a member of the IL-1 family, and is expressed in both extracellular and intracellular pro-inactive protein forms that are activated through proteolytic cleavage. The extracellular form of IL-37 binds the receptor composed of IL-18Rα and IL-1R8 chains, while the intracellular form binds the protein Smad3. Both forms of IL-37 transduce anti-inflammatory signals by suppressing the NF-κB and MAPK pathways while activating the Mer-PTEN-DOK signaling pathways [142]. IL-37 is secreted by innate immune cells and Treg cells [143,144], and plays anti-inflammatory functions in both innate and acquired immune responses by suppressing the activation and production of pro-inflammatory cytokines by monocytes and dendritic cells [145], while also enhancing the immunosuppressive properties of Treg cells [143]. Recent studies have highlighted the pivotal anti-inflammatory role of IL-37 also in the CNS. For instance, the presence of IL-37 in the brain after stroke, observed in both human *post-mortem* tissues and mouse models, is associated with reduced microglial activation [146], suggesting that IL-37 modulates post-stroke inflammation in the brain. Similarly, in the mouse model of Alzheimer’s disease (AD), IL-37 attenuates microglial activation [35]. Moreover, in EAE, IL-37 decreases the percentage of inflammatory infiltrates such as IFN-γ-producing T lymphocytes [147], and promotes the expansion of Treg cells [148].

IL-27, discovered in 2002, is a member of the IL-12 cytokine superfamily. IL-27 is composed of the Epstein–Barr virus-induced gene 3 (EBI3) and p28 chains, and it binds to a heterodimeric receptor formed by IL-27Rα and the signal transducer gp130 chains [149,150]. The engagement of IL-27R recruits Jak kinases, which predominantly activate the STAT1 and STAT3 pathways [151,152]. IL-27-mediated STAT1 activation in CD4 T cells was initially linked to IFN-γ production and Th1 polarization [149,153]. However, in the last few decades, anti-inflammatory functions of IL-27 have also been described, such as the suppression of IL-17 production [154,155] and the induction of IL-10 production by effector T lymphocytes [156].

IL-27 is produced by innate immune cells [157], B lymphocytes [158], Treg cells [159], and a subtype of Breg cells, known as innate-like IL-27-producing natural regulatory B-1a cell (i27-Breg) [160]. Importantly, the expression of IL-27 and its receptor has been detected in the CSF and post-mortem brain tissues of MS patients [161,162]; additionally, several studies have shown that IL-27 dampens inflammation and disease severity in EAE [38,154,156,163].

## 3. Anti-Inflammatory Therapeutic Approaches in ALS

Although evidence supports the anti-inflammatory role of cytokines into CNS, most therapeutic efforts aimed at modifying the neuro-inflammatory response in ALS patients have focused on drugs such as Anakinra (a recombinant analog of the endogenous antagonist IL-1Ra) [164,165], Masitinib (a pluripotent tyrosine kinase inhibitor) [166,167], Tocilizumab (the IL-6 receptor neutralizing antibody) [168], and Ibudilast (an inhibitor of toll like receptor 4 and phosphodiesterase 3 and 4) [169] (Table 1). Preliminary data from trials using these drugs are promising. However, these approaches primarily target inflammatory molecules, and such approaches have a high risk of harm, where toxicity may outweigh a beneficial drug effect. In this context, reinforcing the physiological anti-inflammatory response may provide a more effective approach to control neuroinflammatory states compared to drug-mediated immune suppression. Preclinical studies in this direction have shown exciting potential, and some clinical studies have started in recent years. Notably, infusions of autologous Treg cells in three ALS patients have shown a positive correlation with slowed disease progression according to the Appel ALS scale for each patient [170] (Table 1).

It is well known that interleukin 2 (IL-2) is crucial for the generation, activation and survival of Treg cells [171]. Since low dose IL-2 (ld-IL-2) administration selectively expands Treg cells in both mice and humans [172,173], the effects of ld-IL-2 in ALS was evaluated through a phase-2a, randomized, double-blind, placebo-controlled trial. Results from the trial showed that all individuals in both active treatment groups showed an increase in Treg cells. However, differences in neurofilament light chain protein (Nfl) concentrations, used as surrogate markers of efficacy, were not significantly modulated [174] (Table 1). The main limitation of this study was the small sample size, comprising a highly selected population of slowly progressing patients and a short treatment duration. Interestingly, this trial identified CD27 and Toll like receptor (TLR)9 [175], as well as metabolomic profiles [176], as baseline biomarkers that could predict Treg cell expansion after ld-IL-2 treatment.

Rapamycin, another molecule known to promote Treg cell expansion [177], was tested in a randomized, double-blind, placebo-controlled clinical trial in ALS patients. The results showed that treatment with a low dose of rapamycin is safe in patients with ALS, but failed to demonstrate any significant effect on Treg cells [178,179] (Table 1). Further trials focusing on different outcome measures are necessary to better understand the biological and clinical effects of rapamycin in ALS.

**Table 1 ijms-26-03854-t001:** Therapeutic approaches targeting neuroinflammation in amyotrophic lateral sclerosis (ALS).

Name	Functional Role	Clinical Trial	Phase Trial	Clinical Results	Reference
Anakinra	Antagonist of IL1Ra	NCT01277315	Phase 2	No significant reduction of ALSFRS-R	[165]
Masitinib	Tyrosine kinase inhibitor	NCT02588677	Phase 2/3	Significant decline of ALSFRS-R	[167]
Tocilizumab	Neutralizes IL-6R	-	Pilot study	Decrease inflammation cytokines in PBMC	[168]
Ibudilast	Inhibitor of toll like receptor 4 and phosphodiesterase 3 and 4	NCT02714036	Phase 1	No significant reductions in motor cortical glial activation measured by PBR28-PET SUVR or CNS neuroaxonal loss, measured by serum NfL	[169]
Autologous Treg cells	Induction of suppressive immune response	-	Phase 1	Reduction of Appel ALS scale for each patient	[170]
Low dose IL-2	Expansion of Treg cells	NCT02059759	Phase 2	Significant increase in Treg cells	[174]
Rapamycin	Expansion of Treg cells	-	Pilot study	Treatment is safe	[178]
Rapamycin	Expansion of Treg cells	NCT03359538	Phase 2	Significant increase in Treg cells	[179]

## 4. Conclusions

In recent years, significant advances have been made in understanding neuroinflammation in ALS. It is increasingly recognized that anti-inflammatory cytokines may play a crucial role in regulating neuroinflammation in ALS. This review highlights cytokines with established anti-inflammatory roles, such as IL-4, IL-10, and TGF-β; cytokines with a newly recognized anti-inflammatory role, such as IL-13 and IL-9; and emerging anti-inflammatory cytokines, including IL-25, IL-25, IL-37, and IL-27. We also highlighted the identification of Treg, Breg, M2, Th2, and Th9 cells as both key sources and targets of these cytokines (Figure 2).

In conclusion, enhancing anti-inflammatory cytokines to amplify the regulatory immune response mediated by Treg, Breg, M2, and Th2 cells could represent an innovative therapeutic approach for ALS. While recent studies in this field have yielded encouraging results, several challenges remain, particularly concerning the optimal strategies for amplifying the anti-inflammatory immune response, as well as identifying potential biomarkers to fine-tune personalized therapeutic approaches. Progress in these areas is critical for the future development of new therapies aimed at modulating the immune response for the treatment of ALS and of diseases where neuroinflammation plays a key pathogenic role.

## Figures and Tables

**Figure 1 ijms-26-03854-f001:**
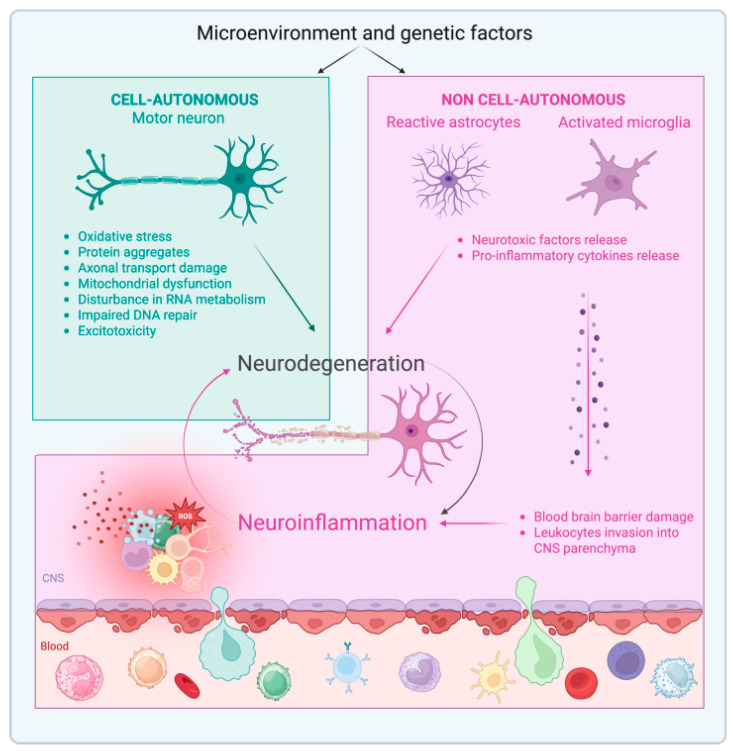
Cell-autonomous and non-cell-autonomous mechanisms of motor neuron degeneration in amyotrophic lateral sclerosis (ALS). Microenvironment and genetic factors might trigger motor neuron intrinsic changes (cell-autonomous mechanism) and several other cell types across the central nervous system (CNS) (non-cell autonomous mechanism). Among non-cell autonomous mechanisms, the most important is the turning of astrocytes and microglia into a pro-inflammatory state. The activation of glia cells in ALS leads to production of neurotoxic factors, which contribute to neurodegeneration, and of pro-inflammatory cytokines involved in the drastic loss of blood-brain barrier (BBB) integrity, which causes the invasion of leukocytes from blood into the CNS parenchyma. The entry of inflammatory monocytes, T lymphocytes, and B lymphocytes into the CNS enhances neuroinflammation by releasing inflammatory cytokines and reactive oxygen species to the tissue. Long and chronic neuroinflammation is linked to impaired tissue function and neurodegeneration, which further amplify inflammatory mechanisms in a pathological feedback loop.

**Figure 2 ijms-26-03854-f002:**
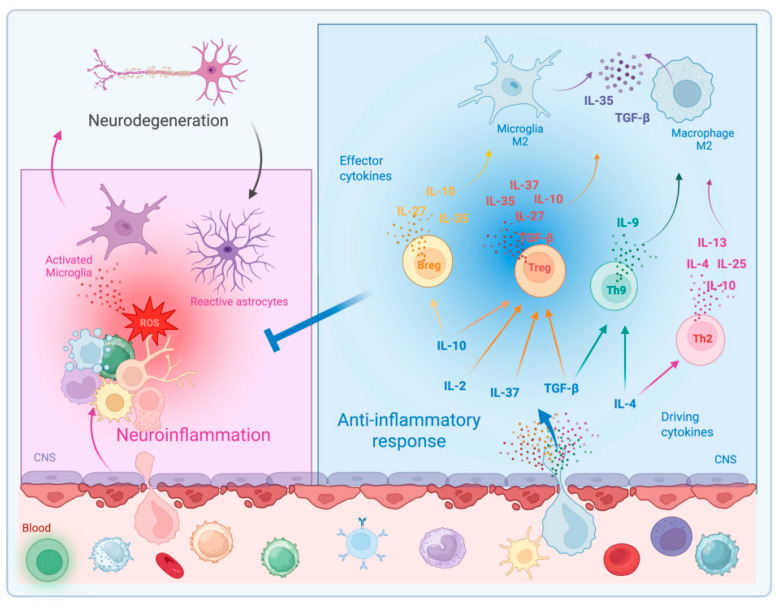
Anti-inflammatory response counteracts the neuroinflammation. In neurodegenerative diseases, the blood-brain barrier (BBB) integrity is impaired, and invasion of leukocytes from blood into CNS parenchyma leads to neuroinflammation. However, immune cells with anti-inflammatory properties might infiltrate the CNS and produce specific anti-inflammatory cytokines contrasting neuroinflammation, and consequently neurodegeneration. In particular, the cytokines involved in this process are IL-10, TGF-β, IL-4, IL-13, IL-9, IL-35, IL-25, IL-37, IL-27, and the immune cells involved are Treg, Breg, microglia M2, macrophage M2, Th2, and Th9 cells. Cytokines may exert either driving than effector functions: IL-4 induces Th2 cells; IL-4 and TGF-β drive Th9 polarization; IL-2, TGF-β, IL-37, and IL-10 are involved in Treg expansion; IL-10 drives Breg; IL-4, IL-10, IL-13, IL-9, IL-25, IL-37, IL-27, and TGF-β polarize microglia and macrophages towards the M2 anti-inflammatory profile.

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
