# Peer review of "Neuroinflammation and Amyotrophic Lateral Sclerosis: Recent Advances in Anti-Inflammatory Cytokines as Therapeutic Strategies"

_ijms, 2025, doi:10.3390/ijms26083854_

Round 1

Reviewer 1 Report

Comments and Suggestions for Authors

I thank the opportunity to review the manuscript entitled “Neuroinflammation and amyotrophic lateral sclerosis: recent advances in anti-inflammatory cytokines as therapeutic strategies” sent for publication in International Journal of Molecular Sciences. The authors presented a well-written review article discussing the main current topics related to knowledge about neuroinflammatory components in Amyotrophic Lateral Sclerosis. Although it represents a more specific theme within the pathophysiology of the disease and is not the predominant component related to the disease, this is a theme that has been gaining much relevance especially in the last 10 years. Several therapeutic proposals based on the inflammatory arm of the pathophysiology of ALS have been attempted in different clinical trials, representing a topic of great interest for a review article. Some points could be evaluated by the authors at this stage of review:  

  1. I suggest the inclusion of a Table summarizing the main data related to the recent approaches related to Phase 2 and 3 studies of anti-inflammatory therapeutic approaches which were evaluated in clinical trials in ALS. The main idea is to include data related to the current phase of studies, the Clinical Trials registry number, and if studies are recruiting or ongoing.
  2. I would like to suggest authors the inclusion of a picture (figure) or a diagram showing the main pathophysiological mechanisms and immune-pathogenesis mechanisms involved with ALS.

Author Response

Reviewer 1:

Comment1: I suggest the inclusion of a Table summarizing the main data related to the recent approaches related to Phase 2 and 3 studies of anti-inflammatory therapeutic approaches which were evaluated in clinical trials in ALS. The main idea is to include data related to the current phase of studies, the Clinical Trials registry number, and if studies are recruiting or ongoing.

Response 1: We thank the Reviewer and we prepared Table 1 describing all anti-inflammatory therapeutic approaches in ALS, including data related to the type of study, the phase of study, the Clinical Trials registry number, and clinical results.

Comment2: I would like to suggest authors the inclusion of a picture (figure) or a diagram showing the main pathophysiological mechanisms and immune-pathogenesis mechanisms involved with ALS.

Response2: We thank the Reviewer for his/her suggestion. However, current Figure 1 already shows the main pathophysiological mechanisms involved with ALS, such as oxidative stress, protein aggregates, axonal transport damage etc., and immune-pathogenesis mechanisms, such as pro-inflammatory cytokines produced by resident glia cells and CNS infiltration by immune cells generating neuroinflammation.

Reviewer 2 Report

Comments and Suggestions for Authors

This review article provides a comprehensive overview of the role of neuroinflammation in the pathogenesis of Amyotrophic Lateral Sclerosis (ALS), with a specific focus on anti-inflammatory cytokines. The authors have described the mechanisms by which neuroinflammation contributes to ALS progression, including glial activation, blood-brain barrier dysfunction, and immune dysregulation. Key anti-inflammatory cytokines have been discussed—such as IL-10, TGF-β, IL-4, IL-13, and IL-9—and highlight emerging candidates like IL-35, IL-25, IL-37, and IL-27, summarizing their immunomodulatory roles and evidence from preclinical and clinical studies. The manuscript is well-organized and detailed. It includes recent and relevant literature, including both clinical and preclinical data, and is certainly timely considering the focus on emerging cytokines in ALS immunotherapy. There are some minor suggestions, for making this article more concise and reader-friendly:

  1. Some places like the mention of TGF-beta signalling description is too detailed, maybe a summary figure would be helpful, and making the content concise. The focus if not the signaling pathway itself, but the role of it in ALS.
  2. IL-14 and IL-13 signaling both result in JAK-STAT activation which does not need to be explained again.
  3. A schematic figure for the roles of these cytokines in ALS would improve the quality of the manuscript.
  4. There are some minor grammatical errors which should be checked again throughout the manuscript.
Comments on the Quality of English Language

It would be better if it is reviewed by a native English speaker. There are minor grammatical errors.

Author Response

Comment1: Some places like the mention of TGF-beta signalling description is too detailed, maybe a summary figure would be helpful, and making the content concise. The focus if not the signaling pathway itself, but the role of it in ALS.

Response1: We agree with Reviewer and we deleted the detailed description of TGF-beta signaling in the revised manuscript. Moreover, we included a summary figure on the role of each cytokine in ALS that is the focus of the review.

Comment2: IL-14 and IL-13 signaling both result in JAK-STAT activation which does not need to be explained again.

Response2: We agree with Reviewer and we deleted the sentence explaining JAK-STAT activation by IL-13 in the revised manuscript.

Comment3: A schematic figure for the roles of these cytokines in ALS would improve the quality of the manuscript.

Response3: We thank the Reviewer and we prepared a schematic figure (Figure 3) illustrating the roles of cytokines in ALS.

Comment4: There are some minor grammatical errors which should be checked again throughout the manuscript.

Response4: We performed another round of proof-reading and further English language editing by a native English speaker.